# Exploration of designing an automatic classifier for questions containing code snippets—A case study of Oracle SQL certification exam questions

Hung-Yi Chen[1], Po-Chou Shih[2]*, Yunsen Wang[3]

**1** Department of Information Management, Chaoyang University of Technology, Taichung, Taiwan,
**2** Department of Industrial Engineering and Management, National Yunlin University of Science and Technology, Yunlin, Taiwan, **3** Department of Accounting & Finance, Feliciano School of Business, Montclair State University, Montclair, New Jersey, United States of America

* pcshih@yuntech.edu.tw

**Data Availability Statement:** The final dataset and accompanying code are available on the Open Science Framework, DOI [10.17605/OSF.IO/83HSA] (https://doi.org/10.17605/OSF.IO/83HSA).

## Abstract

This study uses the Oracle SQL certification exam questions to explore the design of automatic classifiers for exam questions containing code snippets. SQL's question classification assigns a class label in the exam topics to a question. With this classification, questions can be selected from the test bank according to the testing scope to assemble a more suitable test paper. Classifying questions containing code snippets is more challenging than classifying questions with general text descriptions. In this study, we use factorial experiments to identify the effects of the factors of the feature representation scheme and the machine learning method on the performance of the question classifiers. Our experiment results showed the classifier with the TF-IDF scheme and Logistics Regression model performed best in the weighted macro-average AUC and F1 performance indices. The classifier with TF-IDF and Support Vector Machine performed best in weighted macro-average Precision. Moreover, the feature representation scheme was the main factor affecting the classifier's performance, followed by the machine learning method, over all the performance indices.

## Introduction

Question classification associates a class label in the testing criteria with a question. It is a subfield in the document classification. Compared to general documents, questions are shorter, which makes the question classification more challenging [1–3].

The classification of questions in learning assessment is one of the applications of the automatic classifier [2]. When building a test bank, teachers must classify the questions into different exam topics for assembling a test paper with a specific scope according to the testing purpose. For example, Bloom's Taxonomy is a classification of educational objectives to assess students' learning outcomes, including levels of remembering, understanding, applying, analyzing, evaluating, and creating. Questions could be classified using Bloom's Taxonomy to assess the learner's cognitive ability. Alternatively, questions could be classified according to the exam topics to assess the learner's proficiency in specific subjects.

**Funding:** The author(s) received no specific funding for this work.

**Competing interests:** The authors have declared that no competing interests exist.

Classifying questions is more challenging for programming course exams because the question stem or options often contain code snippets in addition to the general text descriptions. The code snippets' particular syntax, such as structure, symbols, and variables, increases the complexity of the question classification. Taking the Oracle SQL certification exam questions as an example, the question stem could describe the report's requirements, and the options provide various SELECT statements. Or, the question stem could be a SQL code snippet, and the question's options could list the descriptions of the code intent. The SQL code snippets contain keywords (such as SELECT, UPDATE, WITH, etc.), field names, table names, and literals. All these make the question classifier design more sophisticated.

Classifying SQL exam questions is to build a test bank for making test papers for learning assessment. Teachers can collect historical questions from different sources and classify them by the exam topic in the test bank. Then, they select questions on specific topics from the test bank when assembling a test paper according to the teaching progress. Classifying questions manually is time-consuming and increases the teaching workload for teachers. An automatic question classifier can help teachers build a test bank effortlessly and reduce the workload.

The Literature on the question classifier in learning assessment primarily focuses on classifying questions with pure text content according to cognitive levels [4, 5] or knowledge domains [6, 7]. However, there needs to be more discussion on the classifier for classifying questions with code snippets, and this study fills the gap in the literature.

The study contributes to teaching by helping teachers automatically classify historical questions to build a test bank. For example, one can apply our study's results to the test bank management module of the automatic test paper assembling system developed by [8]. When importing questions, the system suggests possible topic labels to reduce the classification workload. Then, teachers will have more time to conduct formative or summative assessments to provide students with learning feedback and improve teaching quality.

The study aims to compare various question classifier designs to explore the effects of the feature representation schemes and machine learning methods on the classifier's performance for classifying Oracle SQL certification exam questions. We use a factorial experimental design to identify the effects and the best combination of the feature representation schemes and machine learning methods in the metrics of weighted macro-average Area Under Curve (AUC), weighted macro-average F1 score, and weighted macro-average accuracy.

The rest of the paper is organized as follows. Section two reviews the literature on the question classifiers. Section three presents the research design, including the dataset, the feature representation schemes, the machine learning methods, the search for the optimal parameter for ML methods, the indicators for evaluation, and the experimental design. Section four describes the experiment results and discusses the findings. Finally, the last section concludes the study.

## Literature review

Question classifiers classify a question to a class in a non-overlapping set of classes. It is the multi-label classification if a question has multiple class labels or the single-label classification if only one class label is allowed [9].

Most question classifications applied in the learning assessment belong to the single-label classification problem under the multi-class labels, for example, classifying questions to a cognitive level according to Bloom's Taxonomy [10, 11]. The question length in the learning assessment is generally shorter than the length of other documents. Because of the shorter content, the design of the question classifier is more challenging than the document classifier's [1–3].

**Table 1. The datasets and classification schemes used in the literature to develop question classifiers in various knowledge domains.**

| Dataset ID | Year | Compiler | Knowledge Domain | Num of Items | Classification Framework | Used by Other Studies |
|---|---|---|---|---|---|---|
| DS01 | 2011 | [15] | Multi-disciplines | 272 | Bloom's Taxonomy (BT) 6 classes | [10] |
| DS02 | 2012 | [16] | Multi-disciplines | 600 | BT 6 classes | [12, 13, 17, 18] |
| DS03 | 2012 | [19] | Computer programming | 100 | BT 6 classes | [20] |
| DS04 | 2013 | [21] | Computer science | 654 | BT 6 classes | [11] |
| DS05 | 2017 | [22] | Business and marketing domains | 181 | BT 6 classes | [12] |
| DS06 | 2017 | [23] | Operating systems | 1000 | Revised BT 6 classes | - |
| DS07 | 2017 | [24] | Computer programming | 100 | BT 6 classes | - |
| DS08 | 2017 | [22] | Multi-disciplines | 415 | BT 6 classes | [4, 12] |
| DS09 | 2020 | [13] | Multi-disciplines | 141 | BT 6 classes | - |
| DS10 | 2021 | [10] | Course Learning Outcomes (CLO) statements from course documents | 829 | BT 6 classes | - |
| DS11 | 2021 | [5] | Software engineering | 844 | BT 6 classes | - |

Since the datasets, feature representation schemes, and machine learning methods impact the question classifiers, the remaining is divided into three subsections: Dataset, Feature Representation and Word Weights, and Machine Learning Methods.

## Datasets

Classifier development depends on the dataset and the knowledge domain. A classifier developed from a dataset in one domain is challenging to apply to another [6]. Words in the dataset may have varying weights across classification schemes for various domains [4]. Besides, the class number of the classification scheme and the dataset size affect the classifier's design [11, 12].

The literature's datasets cover several domains, such as business and marketing, computer science, computer programming, and operating systems. Since classification depends highly on the knowledge domain, most studies adopt custom classification schemes [3], such as the schemes for the science subject test [6] or the biomedical exam [7]. There are also general classification schemes, such as Bloom's Taxonomy [13] or Costa Levels of Questioning [14]. Table 1 summarizes the knowledge domains, classification schemes, and the sizes of the datasets from the literature to develop the question classifiers.

## Feature representation and word weights

The question's feature representation is one factor that impacts the classifiers. Previous studies mostly use the Term Frequency-Inverse Document Frequency (TF-IDF) to extract features [3]. Lilleberg et al. [25] pointed out that using the TF-IDF and word embedding schemes (such as Word2Vec) simultaneously performs better than TF-IDF alone because word embedding complements the semantic information that the TF-IDF cannot capture. Mohammed and Omar [13] employed the Term Frequency (TF)-Inverse Document Frequency (IDF) based on Part-Of-Speech (POS), abbreviated as TFPOS-IDF, to determine the word weights. The TFPOS-IDF gives different weights to words based on their POS tags to modify the term frequency in the TF-IDF scheme. Then, Word2Vec is used to extract the semantic, dense features of the words and combine them with their TFPOS-IDF representations. The resultant question feature vectors are dense and can reduce computational complexity and improve learning performance [26].

When words have characteristics that identify a class in the classification scheme, allocating higher weights to these words in the representation scheme can improve the classification performance. When using Bloom's Taxonomy to classify questions, the Enhanced TF-IDF (E-TFIDF) gives verbs more weight, followed by nouns and adjectives [18]. The TFPOS-IDF uses the same concept and further distinguishes the verb types to provide different weights. The TFPOS-IDF performs better than the E-TFIDF and TF-IDF when using the support vector machine [4]. Gani et al. [12] proposed the Enhanced TFPOS-IDF (ETFPOS-IDF), which distinguishes verbs into Bloom's Taxonomy and supporting verbs and gives them different weights. Compared to TF-IDF, the ETFPOS-IDF can improve the accuracy and F1-Measure by 5.2% and 5.7%, respectively.

## Machine learning models

Besides the dataset and the feature representation scheme, the machine learning model affects the classification performance. The support vector machine (SVM) is the most used model in the literature [3]. In most cases, SVM outperforms other models, including the k-Nearest Neighbor (k-NN) [17, 18, 23], Naive Bayes (NB) [17, 18], Linear Regression [13], and Random Forest [12]. But, as an exception case, Abduljabbar and Omar [20] pointed out that k-NN performs better than SVM and NB.

We also found advanced models used in the literature. Osadi et al. [24] proposed an ensemble learning by combining rule-based, SVM, k-NN, and NB models and aggregating these results with the majority voting method to classify questions to cognitive levels in Bloom's Taxonomy. If the dataset is large enough, at least 500 questions, one may consider deep learning models for classifying questions, such as BERT [11], LSTM [10], and CNN [5], because the deep learning models perform better than machine learning models in large datasets. Table 2

**Table 2. The datasets and the machine learning models used in the literature to develop question classifiers.**

| Dataset ID | Year | Feature Rep. | Winning Models | Compared models | Note | Literature |
|---|---|---|---|---|---|---|
| DS01 | 2011 | TF-IDF | Linear SVM | - | Single model | [15] |
| DS02 | 2013 | TF-IDF | SVM | Rule-based, k-NN, NB | - | [17] |
| DS03 | 2015 | Mutual Information | k-NN | SVM, NB | - | [20] |
| DS07 | 2017 | POS for the rule-based method; Word Vector of the WordNet for the other methods. | Ensemble of Rule-based, SVM, k-NN, and NB with the majority voting algorithm and WordNet Similarity | - | - | [24] |
| DS06 | 2017 | - | SVM | k-NNs | Feature representation scheme is not mentioned. | [23] |
| DS02 | 2018 | Enhanced TF-IDF | SVM | k-NN, NB | - | [18] |
| DS02, DS09 | 2020 | TFPOS-IDF and Word Vector of the WordNet | SVM | k-NN, LOGREG | - | [13] |
| DS11 | 2021 | - | CNN | LSTM | - | [5] |
| DS01, DS10 | 2021 | - | LSTM | - | Single model | [10] |
| DS04 | 2021 | - | Google's BERT | - | Single model | [11] |
| DS02, DS05, DS08 | 2022 | ETFPOS-IDF | SVM | ANN, RF | The winning model has the highest average F1 score of three datasets. | [12] |

Abbreviations for machine learning models: SVM: Support Vector Machine; k-NN: k Nearest Neighbors; NB: Naïve Bayes; LOGREG: Logistics Regression; ANN: Artificial Neural Network; RF: Random Forest; LSTM: Long Short-Term Memory; CNN: Convolutional Neural Network

**Table 3. The performance of the winning machine learning models in different datasets in the literature.**

| Dataset ID | Year | Feature Rep. | Winning Model | Accuracy | AUC | Macro F1 | Macro Precision | Macro Recall | Literature |
|---|---|---|---|---|---|---|---|---|---|
| DS01 | 2011 | TF-IDF | Linear SVM | 0.874 | - | 0.393 | 0.858 | 0.291 | [15] |
| DS02 | 2013 | TF-IDF | SVM | 0.923 | - | 0.711 | - | - | [17] |
| DS02 | 2018 | Enhanced TF-IDF | SVM | - | - | 0.860 | 0.866 | 0.861 | [18] |
| DS02 | 2020 | TFPOS-IDF and Word Vector of the WordNet | SVM | - | - | 0.897 | 0.902 | 0.897 | [13] |
| DS02 | 2022 | ETFPOS-IDF | SVM | - | - | 0.840 | - | - | [12] |
| DS03 | 2015 | Mutual Information | k-NN | - | - | 0.898 | - | - | [20] |
| DS04 | 2021 | - | Google's BERT | 0.826 | - | - | - | - | [11] |
| DS05 | 2022 | ETFPOS-IDF | SVM | - | - | 0.748 | - | - | [12] |
| DS06 | 2017 | - | SVM | 0.362 | 0.690 | 0.496 | 0.990 | 0.362 | [23] |
| DS07 | 2017 | POS for the rule-based method; Word Vector of the WordNet for the other methods. | Ensemble of Rule-based, SVM, k-NN, and NB with the majority voting algorithm and WordNet Similarity. | - | - | 0.785 | 0.853 | 0.777 | [24] |
| DS08 | 2022 | ETFPOS-IDF | SVM | - | - | 0.699 | - | - | [12] |
| DS09 | 2020 | TFPOS-IDF and Word Vector | SVM | - | - | 0.837 | 0.856 | 0.844 | [13] |
| DS10 | 2021 | - | LSTM | 0.740 | - | 0.730 | 0.730 | 0.740 | [10] |
| DS11 | 2021 | - | CNN | 0.800 | - | - | - | - | [5] |
| DS01 | 2011 | TF-IDF | Linear SVM | 0.874 | - | 0.393 | 0.858 | 0.291 | [15] |

summarizes the datasets and models used in the literature to develop question classifiers; Table 3 summarizes the performance of the winning models in different datasets.

In summary, the design of the question classifier depends on the knowledge domain and the classification scheme. Previous studies on the question classifier in learning assessment mainly focus on classifying questions with pure text contents. Most studies use TF-IDF to extract features from the questions. Other proposed derived TF-IDF schemes give different weights to words according to the characteristics of the classification scheme to improve the performance. SVM is one of the machine learning models that performs well in most studies. Deep learning models perform better than machine learning models in large datasets. However, there needs to be more discussion on the classifiers that classify questions containing code snippets. This study fills the literature gap by extending the question classifier's design to questions mixed with text and code snippets.

## Methodology

The study adopts the experimental design method to explore the effects of the feature representation schemes and machine learning models on the performance of the question classifiers for questions containing code snippets.

The study has the following assumptions and criteria to classify questions in the dataset:

- The topics in the question classification scheme have basic and advanced inter-dependency. The advanced topics cover the knowledge scope of the basic subjects.

- If a question contains multiple topics, we will label the question as the most advanced topic.

- This study employs the Oracle SQL Expert exam topics as the classification scheme [27]. The exam topics are organized in a two-level hierarchy. Each topic can be further divided into multiple sub-topics. However, due to the limited number of questions in the dataset, only the first-level topics are used in the study. The topics and the number of questions in each

**Table 4. The first-level topics in the Oracle SQL Expert exam and the number of questions in each topic in the dataset.**

| Topic ID | Exam Topic | Count |
|---|---|---|
| 08 | Using Subqueries to Solve Queries | 27 |
| 12 | Use DDL to manage tables and their relationships | 26 |
| 07 | Displaying Data from Multiple Tables | 18 |
| 03 | Restricting and Sorting Data | 18 |
| 05 | Using Conversion Functions and Conditional Expressions | 15 |
| 10 | Managing Tables using DML statements | 13 |
| 06 | Reporting Aggregated Data Using Group Functions | 12 |
| 02 | Retrieving Data using the SQL SELECT Statement | 8 |
| 09 | Using SET Operators | 8 |
| 14 | Controlling User Access | 6 |
| 11 | Managing Indexes Synonyms and Sequences | 6 |
| 04 | Using Single-Row Functions to Customize Output | 6 |
| 13 | Managing Views | 4 |
| 01 | Relational Database concepts | 4 |

topic are shown in Table 4. In the table, higher topic numbers indicate more advanced topics.

## Dataset

The dataset contains 171 questions written in English and formatted with Markdown syntax and symbols. The question content may contain general text descriptions, markdown symbols, and SQL code snippets. The code snippets may appear in the question stem or the answer options.

The text in the SQL code snippets can be further divided into several types:

- SQL reserved words, e.g., CREATE TABLE, SELECT, INSERT, UPDATE, etc.

- Literals, e.g., numbers, strings, date strings, etc.

- Data type keywords

- Operators

- Pseudo-columns, e.g., NEXTVAL

- Schema object names, e.g., names for the table, column, function, sequence, etc.

Questions in the dataset were annotated with the topic identifications according to the assumptions and criteria mentioned before. Table 4 summarizes the categorization. The topic, "Using Subqueries to Solve Queries," contains the most questions, a total of 27 items accounting for 15%. The two topics, "Managing Views" and "Relational Database concepts," all contain the least number of questions; each has four items, accounting for 2%. The dataset is highly unbalanced. The topic with the most questions is about six times more than the least.

## Data preprocessing

The data are pre-processed before extracting the features according to the following steps:

1. Lowercase conversion and removal of special symbols: Convert all texts to lowercase. Remove the Markdown syntax, such as the image syntax "!()[]" and the table syntax "—+ —," etc. However, the SQL operators, such as +, -, *, /, -, %, _, etc., are retained.

2. Lemmatization: The study used the spaCy en_core_web_sm 3.6.0 package [28] for lemmatization.

3. Revise the incorrect lemmatization results: For example, the column name "ord_no" is lemmatized to "ord _ no," which loses the original meaning.

4. Stop word removal: Remove the stop words from the content, but do not include the SQL keywords. This study used the NLTK package [29] to handle the stop words.

The study does not perform stemming in the pre-processing process because it makes words lose their original meanings and affects the classification performance. For example, the SELECT in the SQL statement and the "selecting" and "selection" in the general text will all be converted to "select" after stemming, which makes the context information lost.

## Feature representation

The study considers three feature representation schemes commonly used in the literature: Term Frequency-Inverse Document Frequency (TF-IDF), Word2Vector, and FastText.

TF-IDF value indicates the importance of a word in a document. When a word appears only in a single document and has a high frequency, the word is an essential feature of the document. The TF-IDF value of word $i$ in question $j$ is $f_{ij} = tf_{ij} \times idf_i$, where $tf_{ij}$ is the frequency of word $i$ in question $j$, and $idf_i$ is the scarcity measure to the word across all documents. The higher the word scarcity, the larger value of $idf_i$.

The process of using TF-IDF to vectorize a question involves two steps. In the fitting step, the TF-IDF model is trained on the training dataset to build the vocabulary and calculate $idf_i$ for each word in the vocabulary. Then, in the transform step, the learned vocabulary and the $idf_i$ values are used to encode a new question to a vector. The word frequency in the new question is multiplied by each word's $idf_i$ value to generate the question vector. Specifically, let $V$ be the vocabulary. Then, the vector of the new question $j'$ is:

$$\overrightarrow{v_{j'}} = [tf_{1j'}, \ldots, tf_{ij'}, \ldots, tf_{|V|j'}) \times [idf_1, \ldots, idf_i, \ldots, idf_{|V|}]^T, \tag{1}$$

where $tf_{ij}$, is the frequency of word $i$ in new question $j'$ and $i \in V$.

The second considered scheme, Word2Vec, is a word embedding model in natural language processing. Word2Vec represents a word with a fixed-dimension vector. Two adjacent word vectors mean they have similar meanings. Compared with the TF-IDF scheme, Word2Vec can capture the word's semantic meaning. The study used Google's 300-dimensional pretrained word embeddings [30].

The last scheme, FastText, is also a word embedding model in natural language processing [31]. Instead of word vectors, FastText generates the embeddings of the n-gram characters that compose the word. When encountering an unknown word in the training dataset, FastText can split the word into multiple characters and combine the embeddings of these characters to generate the word embedding, which overcomes the Out-of-Vocabulary (OOV) problem.

We generate a question embedding by averaging the word embeddings for the question. Let $\overrightarrow{v_i} = [e_1, \ldots, e_k, \ldots, e_m]$ denote the embedding vector for word $i$, where $e_k$ is the $k$-th element in the embedding vector, and $m$ is the embedding dimension. Additionally, let $V_j$ be the set of words in question $j$. Then, the embedding vector for question $j$ is:

$$\overrightarrow{v_j} = \sum_{i \in V_j} \overrightarrow{v_i} \times \frac{1}{|V_j|} \tag{2}$$

## Machine learning models

The study considers four commonly used models in the literature: Multinominal Naive Bayes (MNB), Logistics Regression (LOGREG), Linear Support Vector Machine (LSVM), and Support Vector Machine (SVM). The following describes the characteristics and parameters of each method.

MNB model has a high learning bias and is suitable for a small amount of data, commonly used as a baseline for comparison in the literature [32]. MNB's parameter $\alpha \geq 0$, a pseudo-count value, smooths the likelihood of a word in a class to avoid the zero-probability problem. The parameter makes the words not in the training samples have a non-zero probability of preventing zero probability in further computations. A more considerable $\alpha$ value results in a more significant smoothing effect and a simpler model.

LOGREG performs well in high-dimensional, sparse data. The parameter C controls the regularization strength when using L2 regularization in the LOGREG. A larger C value results in a smaller regularization strength and a more complex model [33].

LSVM is the most used method in the literature and performs very well [3]. Like the parameter in LOGREG, the parameter C controls the regularization strength when using L2 regularization in the LSVM.

Besides the linear classifiers of LOGREG and LSVM, the study also considers the non-linear classifier SVM adopting the Radial Basis Function (RBF) kernel. The RBF kernel has two parameters: C and $\gamma$. The parameter C controls the regularization strength to control the model complexity, as mentioned in the previous two classifiers. The parameter $\gamma$ controls the influence of the training observations near the decision boundary since the observations near it determine the decision boundary. The larger the $\gamma$ value is, the smaller the impact of the observations far from the decision boundary is, which has a smoother decision boundary [34].

## Parameter optimization

This study employs stratified K-fold cross-validation to evaluate the generalization ability of the models in the learning phase. The stratified split of the training dataset ensures that each fold contains learning instances in all classes, which is suitable for the unbalanced dataset.

In our dataset, the 01 and 13 topics contain the fewest questions, with only four questions each. The stratified split divides the dataset into the training and test datasets. In the training dataset, the 01 and 13 topics contain only three questions each. Therefore, the study uses three-folds in the cross-validation to identify the model parameters with the best generalization ability.

This study uses the grid search to identify the model parameters to make a model with good generalization ability. The grid search performs cross-validation for each parameter combination and selects the best combination. The parameter search points for each model are set as follows:

- MNB parameter $\alpha$: logarithm scale interval $[\log_{10}-2, \log_{10}4]$, 20 equal parts.

- C parameter in LOGREG, LSVM, and SVM: logarithm scale interval $[\log_{10}-2, \log_{10}10]$, 20 equal parts.

- $\gamma$ parameter in SVM: logarithm scale interval $[\log_{10}-9, \log_{10}3]$, 13 equal parts.

When the grid search algorithm searches the best parameters, the study uses the One-versus-Rest (OvR) strategy to convert a multi-class into a binary classification problem. The study uses the Area Under the Curve (AUC) of the Receiver Operating Characteristic (ROC) to guide the search, as AUC is a simple and robust metric for comparing the effectiveness of classification models.

## Evaluation metrics

The study measures the classifier performance with the following metrics: weighted precision, weighted F1-score, and weighted AUC because of the unbalanced question dataset in the study. The weighted metrics consider the class imbalance by weighting the class metrics by their proportion. Their metrics are presented as the following.

The recall rate measures the ability of the classifier to detect positive instances, given all positive instances that exist either in the true positive or false negative cases. Let TP be the number of true positive instances, FP be the number of false positive instances, TN be the number of true negative instances, and FN be the number of false negative instances, respectively. Under the One-vs-Rest (OvR) multiclass strategy, the recall rate for the class $c$ is calculated as follows:

$$R_c = \frac{TP_c}{TP_c + FN_c}. \tag{3}$$

Let $w_c$ be the proportion of the class $c$ in the dataset. Then, the weighted macro-average recall rate for all classes is:

$$wR = \sum_{c=1}^{L} w_c R_c, \tag{4}$$

where L is the total classes in the dataset.

The precision rate measures the ability of the classifier to identify positive instances given all predicted positive instances composed of TP and FP cases. The precision rate of the class $c$ is:

$$P_c = \frac{TP_c}{TP_c + FP_c}. \tag{5}$$

Then, the weighted macro-average precision rate for all classes is:

$$wP = \sum_{c=1}^{L} w_c P_c. \tag{6}$$

The F1 score is the harmonic mean of the precision and recall rates, considering the balance between the precision and recall rates. When the precision and recall rates are not equal, the F1 score decreases. The F1 score of the class $c$ is:

$$F1_c = 2 * \frac{P_c * R_c}{P_c + R_c}. \tag{7}$$

Then, the weighted macro-average F1 score for all classes is:

$$wF1 = \sum_{c=1}^{L} w_c F1_c. \tag{8}$$

The ROC curve is not affected by the imbalanced dataset and is suitable for evaluating classifiers. Plotting an ROC curve requires the true positive rate (TPR) and the false positive rate (FPR) as the X and Y axes, respectively. The TPR is the same as the recall rate. The area under the ROC curve (AUC) quantifies the classifier performance. A large AUC value indicates the TPR is higher than the FPR, which means the classifier performs better. A random guess classifier has an AUC value of 0.5, and a perfect classifier achieves 1.

The study employs the weighted macro-average FPR and TPR to calculate the AUC value of the model. The FPR of the class c is:

$$FPR_c = \frac{FP_c}{FP_c + TN_c}. \tag{9}$$

Then, the weighted macro-average FPR for all classes is:

$$\text{wFPR} = \sum\nolimits_{c=1}^{C} w_c \text{FPR}_c. \tag{10}$$

Since the TPR is the same as the recall rate, the weighted macro-average TPR for all classes is the same as Eq (4).

The accuracy rate is biased in the unbalanced dataset because it is affected by the majority class [35]. Since the question dataset in the study is unbalanced, the study does not employ the accuracy rate for evaluation. Additionally, the study does not adopt the weighted macro-average recall rate since it equals to the accuracy rate.

## Experiment design and data analysis procedure

**Factors and responses.**   The study employs factorial experiments to examine the effects of feature representation schemes and machine learning models on the classifier's performance. The experiment considers two factors and three responses, as shown in Table 5. The first factor is the feature representation scheme, which has three levels: TF-IDF, Word2Vec, and FastText.

The second factor is the machine learning model, which has four levels: Logistics Regression, Multi-nominal Naive Bayes, Linear Support Vector Machine, and Support Vector Machine with the Radial Basis Function as the kernel.

The response variables include weighted macro-average AUC, weighted macro-average precision, and weighted macro-average F1 score.

The two factors result in 12 groups. The experiment replicates 300 trials for each group, resulting in 3600 trials in total.

**Power analysis.**   This study conducts the power analysis to ensure the statistical significance of the experiment results using the G*Power software [36]. The power of the experiment is 1.000 under the following conditions: the type I error $\alpha = 0.05$, the degrees of freedom of the factors $(3-1) \times (4-1) = 6$, the sample size 3600, and the effect size $\eta^2 = 0.025$.

**Program implementation and data analysis tools.**   The study implements the classifiers and the grid search with the Scikit-Learn library [37]. R language and related packages are used for data analysis and plotting.

**Analysis procedure and methods.**   The data analysis procedure is as follows:

1. Descriptive statistics and data distribution analysis: Analyze the mean, standard deviation, skewness, and kurtosis of the three responses. The normal distribution and homogeneity of variance across groups are tested.

**Table 5. The factors and responses in the experiment.**

| Independent Variables | | Response Variables |
|---|---|---|
| **Factors** | **Levels** | |
| Feature Representation Schemes (FRS) | TF-IDF: Term Frequency—Inverse Document Frequency | wAUC: Weighted macro-average AUC. |
| | Word2Vec: Google News pre-trained word embeddings with 300 dimensions | wP: Weighted macro-average precision. |
| | FastText: Facebook pre-trained word embeddings with 300 dimensions | wF1: Weighted macro-average F1. |
| Machine Learning Models (MLM) | LOGREG: Logistics Regression | |
| | MNB: Multinomial Naïve Bayes | |
| | LSVM: Linear Support Vector Machine | |
| | SVM: Support Vector Machine with Radial Basis Function (RBF) kernel | |

2. Analysis of Variance: The parametric ANOVA is performed if the response variable meets the normal distribution and homogeneity of variance assumptions. Otherwise, the non-parametric Aligned Ranks Transformation ANOVA (ART ANOVA) is used. ART ANOVA, a non-parametric method, does not require the data to meet the above two assumptions. ART ANOVA can analyze the main and interaction effects [38, 39], which is more suitable than the Sheirer-Ray-Hare Test [40].

3. Post-hoc Analysis: If the response variable meets the assumptions of the normal distribution and homogeneity of variance, the Tukey HSD post-hoc analysis is performed. Otherwise, the Aligned Ranks Transformation Contrast is used to find the differences and effect sizes between the levels of each factor.

4. Effect Size measurements in ANOVA and t-test: $\eta^2$ is employed to measure the effect size of a significant effect in ANOVA. The thresholds for small, medium, and large effect sizes are 0.01, 0.06, and 0.14, respectively. Cohen's d measures the effect size of the t-test. The thresholds for small, medium, and large effect sizes are 0.2, 0.5, and 0.8, respectively.

For the concise, the nomenclatures in the S1 Appendix are used to express the statistical results.

## Experiment results and discussion

### Experiment results

Table 6 summarizes the experiment results for the three performance metrics. The factors FRS, MLM, and their interaction significantly affect the weight macro-average AUC with large effects. The classifier using the TF-IDF scheme with the LOGREG model achieves the best performance, with a mean of 93.812%. Next, in the metric of the weighted macro-average precision, FRS and MLM factors significantly affect the performance with large effect sizes and their interaction with the intermediate effect. The classifier with the TF-IDF schema and the SVM model performs best, with a mean of 72.574% on the weighted macro-average precision. Lastly, the factors FRS, MLM, and their interaction significantly affect the weighted macro-average F1-score with large effect sizes. The classifier using the TF-IDF scheme and the LOG-REG model achieves the best mean, averaging 81.574% on the weighted macro-average F1-score. The following subsections present the details.

**Table 6. Summary of the experiment results for the three performance metrics: weighted macro-average AUC (wAUC), weighted macro-average precision (wP), and weighted macro-average F1-score (wF1).**

| Effects | F-Value | Effect Size ($\eta^2$) | Magitude of Effec Size | Best Level | Mean of Metric |
|---|---|---|---|---|---|
| | | | Metric: wAUC(%) | | |
| FRS | 3423.550 | 0.656 | Large | TF-IDF | 92.924 |
| MLM | 757.710 | 0.388 | Large | LOGREG | 88.900 |
| FRS:MLM | 343.700 | 0.365 | Large | (TF-IDF, LOGREG) | 93.812 |
| | | | Metric: wP(%) | | |
| FRS | 774.470 | 0.302 | Large | TF-IDF | 72.378 |
| MLM | 110.700 | 0.156 | Large | LSVM | 70.632 |
| FRS:MLM | 191.160 | 0.138 | Intermediate | (TF-IDF, SVM) | 72.574 |
| | | | Metric: wF1(%) | | |
| FRS | 2329.630 | 0.565 | Large | TF-IDF | 61.468 |
| MLM | 1320.740 | 0.525 | Large | LOGREG | 57.929 |
| FRS:MLM | 217.990 | 0.267 | Large | (TF-IDF, LOGREG) | 63.545 |

**Weighted macro-average AUC (wAUC).** The mean, median, mode, standard deviation, skewness, and kurtosis of all wAUC values are 86.581%, 87.092%, 86.093%, 6.31, -0.699, and 0.273, respectively. The wAUC values do not distribute as the normal curve (Anderson-Darling Stat = 27.489 $p$ <0.001) and do not have homogeneity of variance (Levene's Test $F$ = 33.844 < 2.2e-16); Fig 1 shows the wAUC distributions of the twelve groups. The groups in the TF-IDF levels have smaller variances than the others.

Aligned Ranks Transformation ANOVA (ART ANOVA) was used to analyze the variances since the wAUC values do not meet the normality and homogeneity of variance assumptions.

The FRS factor ($F$ = 3423.55, $p < 2.22e-16$, DF = 2), and the MLM factor ($F$ = 757.71, $p < 2.22e-16$, DF = 3) were significant; so was the interaction ($F$ = 343.70, $p < 2.22e-16$, DF = 6). The FRS factor generated the largest effect size ($\eta^2_{FRS}$ = 0.656), followed by the MLM factor ($\eta^2_{MLM}$ = 0.388) and the interaction ($\eta^2_{FRS:MLM}$ = 0.365).

The means for the factors and their interaction are shown in Fig 2. In the FRS factor, the TF-IDF level performed the best ($m_T$ = 92.924), and the worst was the FastText level ($m_F$ = 82.934). The post-hoc analysis indicated that the TF-IDF produced a significantly larger effect size than the other two levels ($t_{T-W}$ = 67.404, p < 0.0001, $d_{T-W}$ = 3.05; $t_{T-F}$ = 75.269, p < 0.0001, $d_{T-F}$ = 2.09).

For the MLM factor, LOGREG had the best mean ($m_L$ = 88.900), and MNB had the worst ($m_M$ = 82.671), as shown in Fig 2. The post-hoc analysis indicated that the LOGREG produced a significantly large effect size compared to the MNB ($t_{L-M}$ = 44.320, $p < 0.0001$, $d_{L-M}$ = 0.909). Although the mean of the LOGREG was greater than that of the LSVM, the effect size

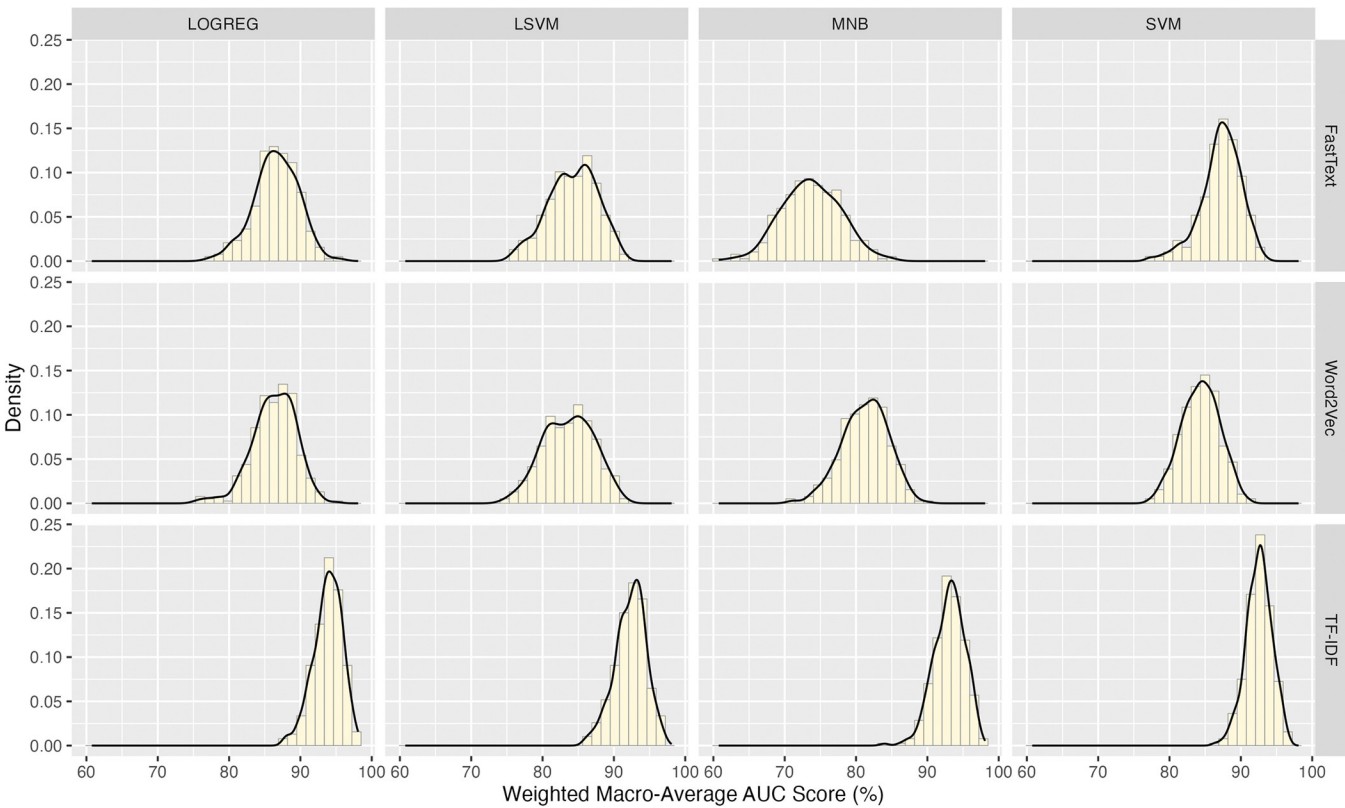

**Fig 1. The distributions of the weighted macro-average AUC values for groups of various combinations of feature representation schemes and machine learning models.**

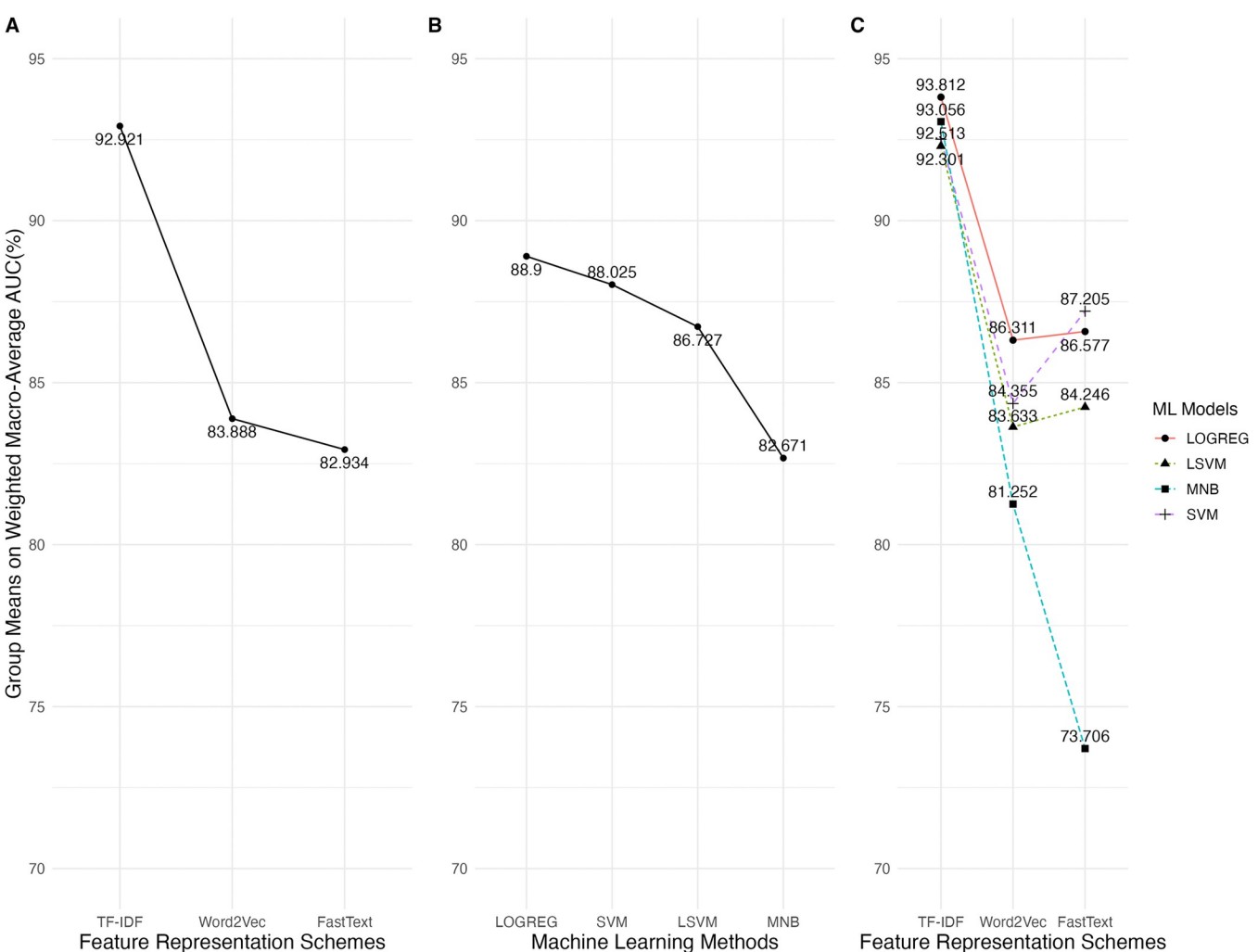

**Fig 2. The mean analysis of the weighted macro-average AUC values for FRS, MLM factors, and their interactions.**

was significantly small ($t_{L-LS}$ = 17.439, $p < 0.0001$, $d_{L-LS}$ = 0.459). Likewise, the effect size between the LOGREG and SVM was significantly small ($t_{L-S}$ = 6.944, $p < 0.0001$, $d_{L-S}$ = 0.203).

For the interaction of FRS and MLM factors, the group with the best mean was (TF-IDF, LOGREG) with $m_{(T,L)}$ = 93.812, followed by (TF-IDF, MNB) with $m_{(T,M)}$ = 93.056, as shown in Fig 2. However, no significant difference existed between the two groups ($t_{(T,L)-(T,M)}$ = 3.597, $p$ = 0.0170). The mean of the group (TF-IDF, LSVM) ($m_{(T,LS)}$ = 92.3) was close to that of the group (TF-IDF, SVM) ($m_{(T,S)}$ = 92.5), and no significant difference existed between the two groups ($t_{(T,LS)-(T,S)}$ = -1.062, $p$ = 0.9961). The group with the worst mean was (FastText, MNB) with $m_{(F,M)}$ = 73.706. The effect size between the best and worst groups was significantly large ($t_{(T,L)-(F,M)}$ = 72.962, $p < 0.0001$, $d_{(T,L)-(F,M)}$ = 5.07).

Since the variance of wAUC values did not homogeneous across groups, the post-hoc analysis examined the effect sizes in the four quantiles, as shown in Fig 3. In Fig 3, The best group (TF-IDF, LOGREG) is set to be the reference group. The three horizontal dash lines denote the thresholds for the small, medium, and large effect sizes. The reference group has the best performance in all quantiles. The differences caused by the best group ranged from small to large effect sizes compared to the other groups.

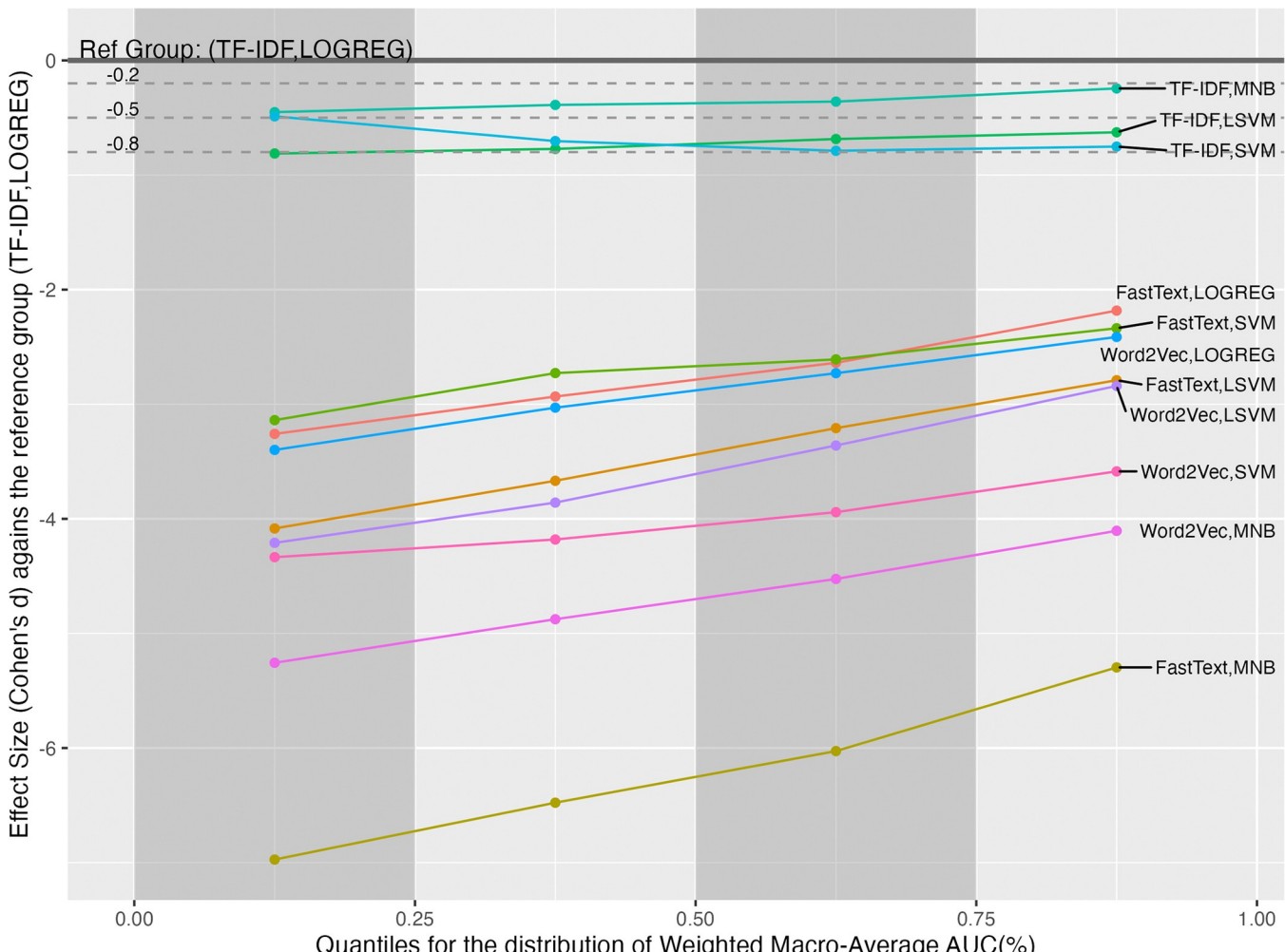

**Fig 3. The effect size analysis on the weighted macro-average AUC values in the four quantiles for the interactions between the FRS and MLM factors.**

**Weighted macro-average precision (wP).** The mean, median, mode, standard deviation, skewness, and kurtosis of all wP values are 66.284%, 67.232%, 70.172%, 9.553, -0.470, and -0.035, respectively. The wP values do not distribute as the normal curve (Anderson-Darling Stat = 13.4768, $p < 0.001$) and do not have homogeneity of variance (Levene's Test $F = 38.273$, $p < 2.2e\text{-}16$). Fig 4 shows the wP distributions in all factor-level combinations. The data in the group (Word2Vec, MNB) is distributed as a bimodal curve. The groups with the TF-IDF scheme exhibit less variance than the others.

This study uses the non-parametric ART ANOVA to analyze the variances of the wP values. The FRS factor produced a significantly large effect size ($F = 774.47$, $p < 2.22e\text{-}16$, DF = 2, $\eta^2_{FRS}$ = 0.302); So did the interaction of the FRS and MLM factors ($F = 110.70$, $p < 2.22e\text{-}16$, DF = 6, $\eta^2_{FRS:MLM}$ = 0.156). However, the MLM factor produced a significantly medium effect size ($F = 191.16$, $p < 2.22e\text{-}16$, DF = 3, $\eta^2_{MLM}$ = 0.138).

Fig 5 shows the mean analysis of the wP values for the FRS, MLM factors, and their interaction. In the FRS factor, the TF-IDF level contributed the best mean ($m_T = 72.378$), and the worst was the FastText level ($m_F = 61.895$). The post-hoc analysis indicated that the TF-IDF produced a significantly larger effect size than the FastText ($t_{T-F} = 37.769$, $p < 0.0001$, $d_{T-F} = 1.33$).

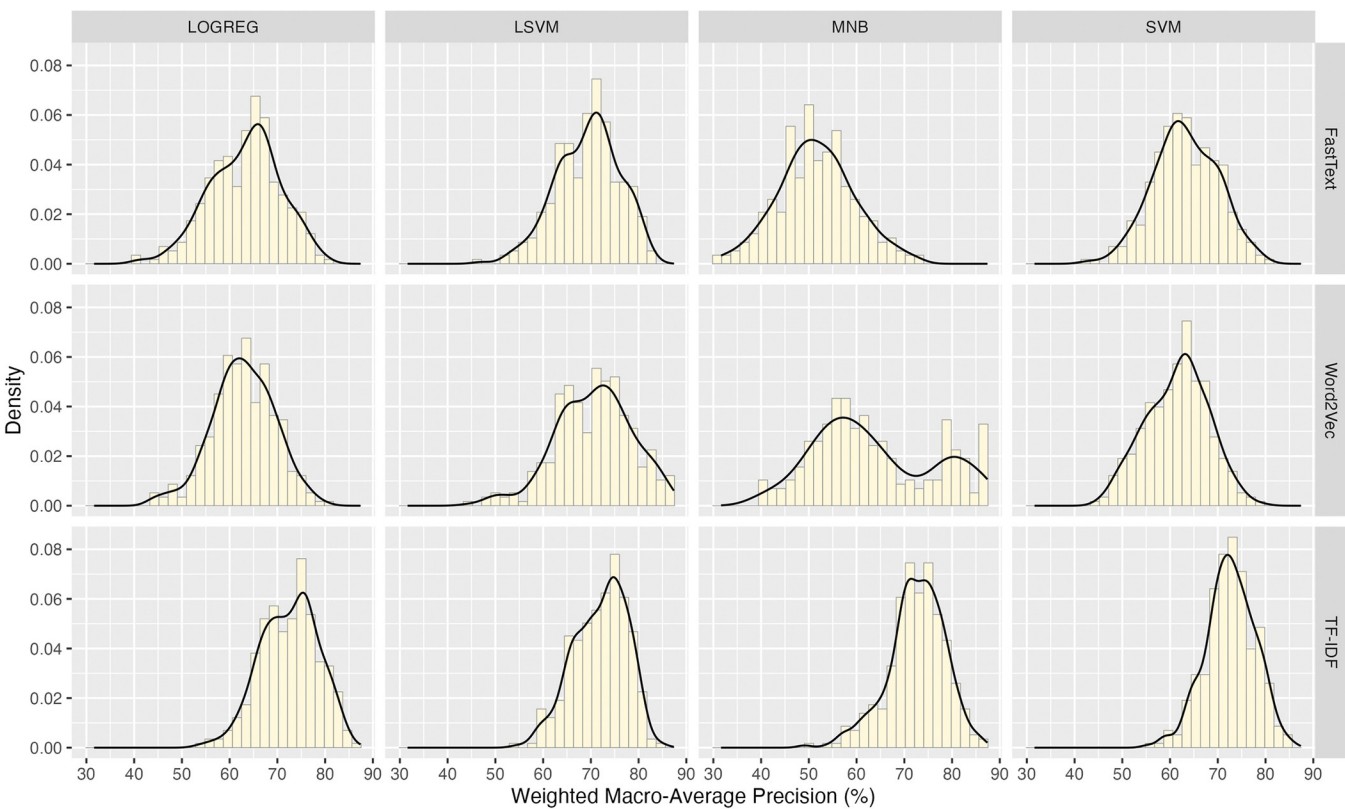

**Fig 4. The distributions of the weighted macro-average precision values for groups of various combinations of feature representation schemes and machine learning models.**

For the MLM factor, the LSVM level contributed the best mean ($m_{LS} = 70.632$), and the worst was the MNB level ($m_M = 62.562$), as shown in Fig 5. The post-hoc analysis indicated that their difference was significantly medium ($t_{LS-M} = 23.672$, $p < 0.0001$, $d_{LS-M} = 0.795$). Instead, the means of the LOGREG and SVM levels were close ($m_L = 66.184$, $m_S = 76.76$), and no significant difference existed between the two levels ($t_{L-S} = 1.565$, $p = 0.3988$).

As for the interactions, the groups with the TF-IDF scheme but different machine learning models performed quite closely. There were no significant differences between the groups. The means for the various machine learning models from high to low were SVM ($m_{(T,S)} = 72.574$), MNB ($m_{(T,M)} = 72.515$), LOGREG ($m_{(T,L)} = 72.484$), and LSVM ($m_{(T,LS)} = 71.939$), given the TF-IDF scheme.

The worst group was (FastText, MNB) with $m_{(F,M)} = 51.584$. Compared to the worst group, the best group (TF-IDF, SVM) generated a significantly large effect size ($t_{(T,S)-(F,M)} = 31.713$, $p < 0.0001$, $d_{(T,S)-(F,M)} = 3.11$).

As the variance of wP values was not evenly distributed among the groups, the post-hoc analysis examined the effect sizes within each quantile, as shown in Fig 6. The reference group is (TF-IDF, SVM) with the best mean of wP values among all groups. The reference group performed the best in the first two quantiles. But, in the third and fourth quantiles, the best groups become (TF-IDF, LOGREG) and (Word2Vec, MNB) respectively. In the fourth quantile, the group's mean (Word2Vec, MNB) was greater than that of the reference group, and the effect size was significantly small ($d_{(W,M)-(T,S)}^{[4]} = 0.264$).

**Weighted macro-average F1 (wF1).** The mean, median, mode, standard deviation, skewness, and kurtosis of all wF1 values are 50.829%, 53.342%, 58.216%, 12.796, -0.672, and 0.141,

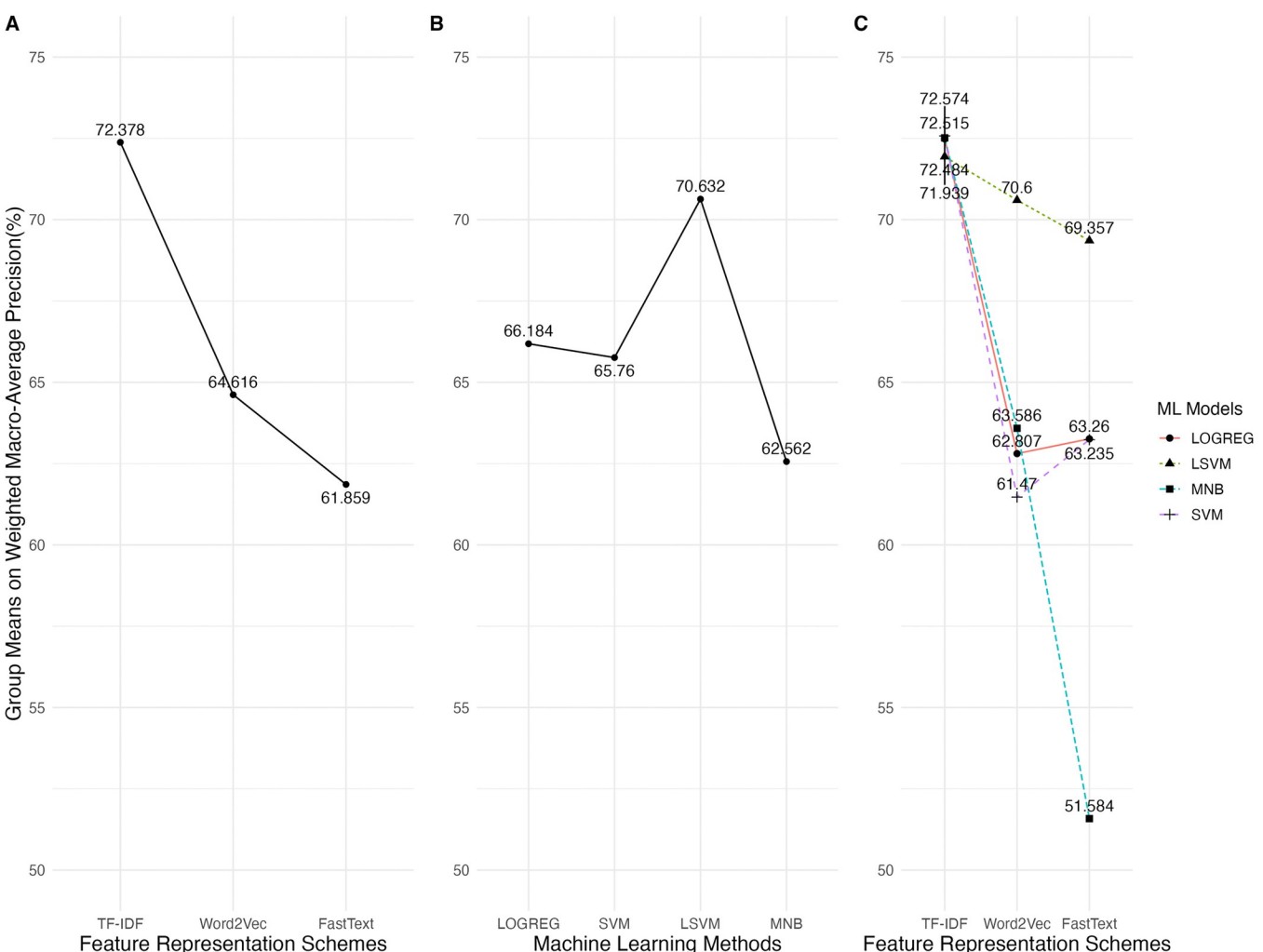

**Fig 5. The mean analysis of the weighted macro-average precision values for FRS, MLM factors, and their interactions.**

respectively. The wF1 values did not distribute as the normal curve (Anderson-Darling Stat = 37.477 p < 0.001) and were not evenly distributed among the groups (Levene's Test $F$ = 29.637 < 2.2e-16). Fig 7 shows the wF1 distributions in all factor-level combinations. A bimodal shape occurred in the group (Word2Vec, MNB). And, like the results in the wAUC and wP, the groups with the TF-IDF scheme caused less variance than the others.

This study uses non-parametric ART ANOVA to analyze the variances since the distribution of wF1 values did not meet the assumptions for the parametric ANOVA. The FRS factor ($F$ = 2329.63, $p$ < 2.22e-16, DF = 2), the MLM factor ($F$ = 1320.74, $p$ < 2.22e-16, DF = 3), and their interaction ($F$ = 217.99, $p$ < 2.22e-16, DF = 6) were all statistically significant. The effect sizes of the FRS, MLM, and their interaction were large ($\eta^2_{FRS}$ = 0.565, $\eta^2_{MLM}$ = 0.525, $\eta^2_{FRS:MLM}$ = 0.267).

The mean analysis of the factors and their interactions are shown in Fig 8. In the FRS factor, the TF-IDF level contributed the best mean ($m_T$ = 61.468), followed by the FastText level ($m_F$ = 45.396), and the worst was the Word2Vec level ($m_W$ = 45.623). As indicated by the post-hoc analysis, the TF-IDF generated a significantly larger effect size than the FastText ($t_{T-F}$ =

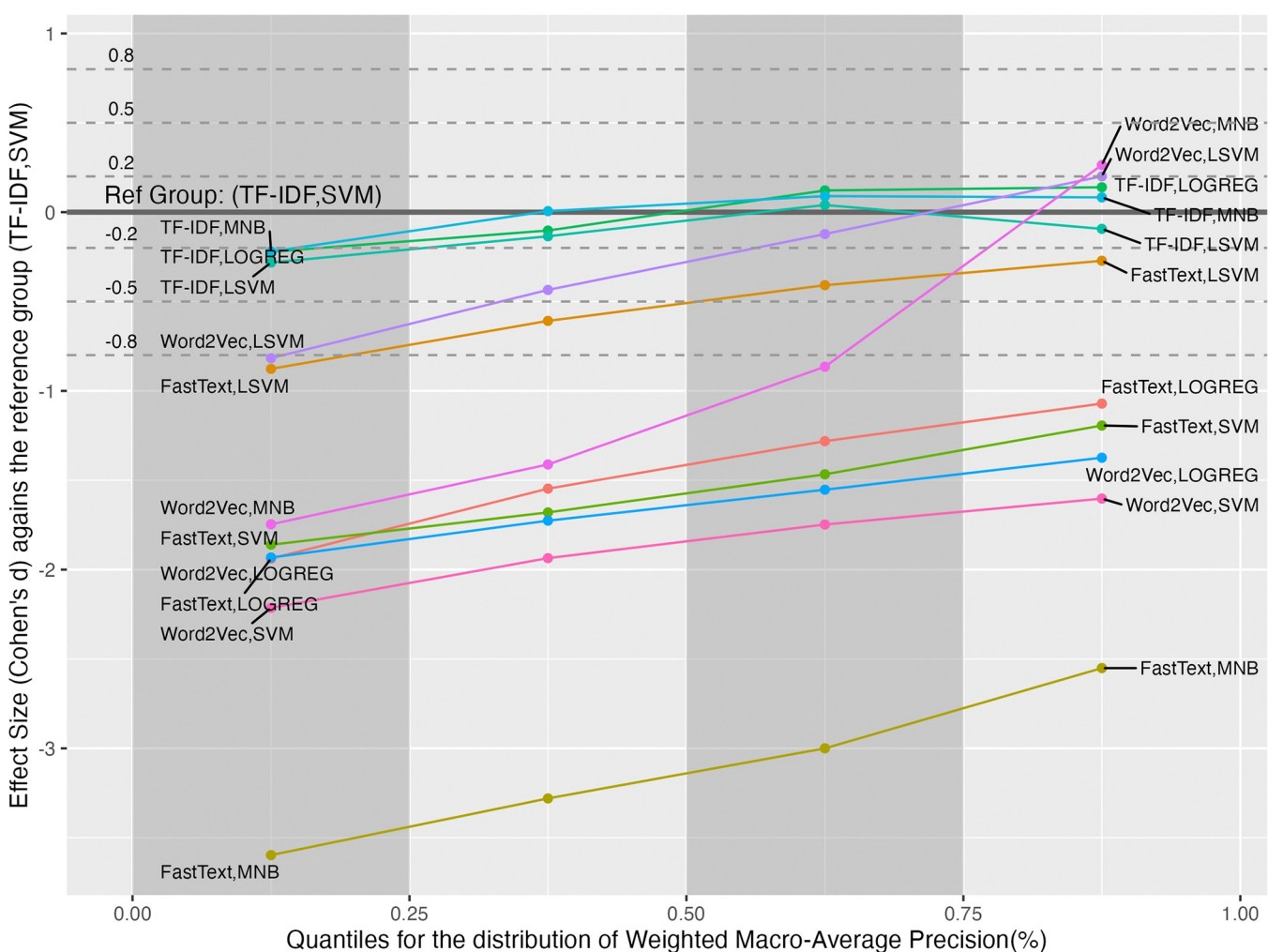

**Fig 6. The effect size analysis on the weighted macro-average precision values in the four quantiles for the interactions between the FRS and MLM factors.**

59.649, $p < 0.0001$, $d_{T-F} = 1.72$). Nevertheless, no significant difference existed between the Word2Vec and FastText levels.

As for the MLM factor, the best and worst levels were LOGREG ($m_L = 57.929$) and MNB ($m_M = 41.938$), respectively. Their difference in effect size was significantly large ($t_{L-M} = 52.077$, p < 0.0001, $d_{L-M} = 1.330$).

Fig 8 also shows the mean analysis of the interactions between the FRS and MLM factors. The best group was (TF-IDF, LOGREG) with $m_{T,L} = 63.545$, and the worst was (FastText, MNB) with $m_{F,M} = 32.71$. Additionally, their differences in effect size were significantly large ($t_{(T,L)-(F,M)} = 54.802$, p < 0.0001, $d_{(T,L)-(F,M)} = 4.120$).

Due to the uneven variance of wF1 values across the groups, the study examined the effect sizes within four quantiles, respectively, as shown in Fig 9. The figure uses the group (TF-IDF, LOGREG) with the best mean as the reference for comparing effect sizes. Unlike the case in the wP, the best group remained the same in all quantiles. Note that the effect sizes of all groups were almost flat from the first to fourth quantiles, except the group (Word2Vec, MNB). The exceptional group's effect size increased after the second quantile.

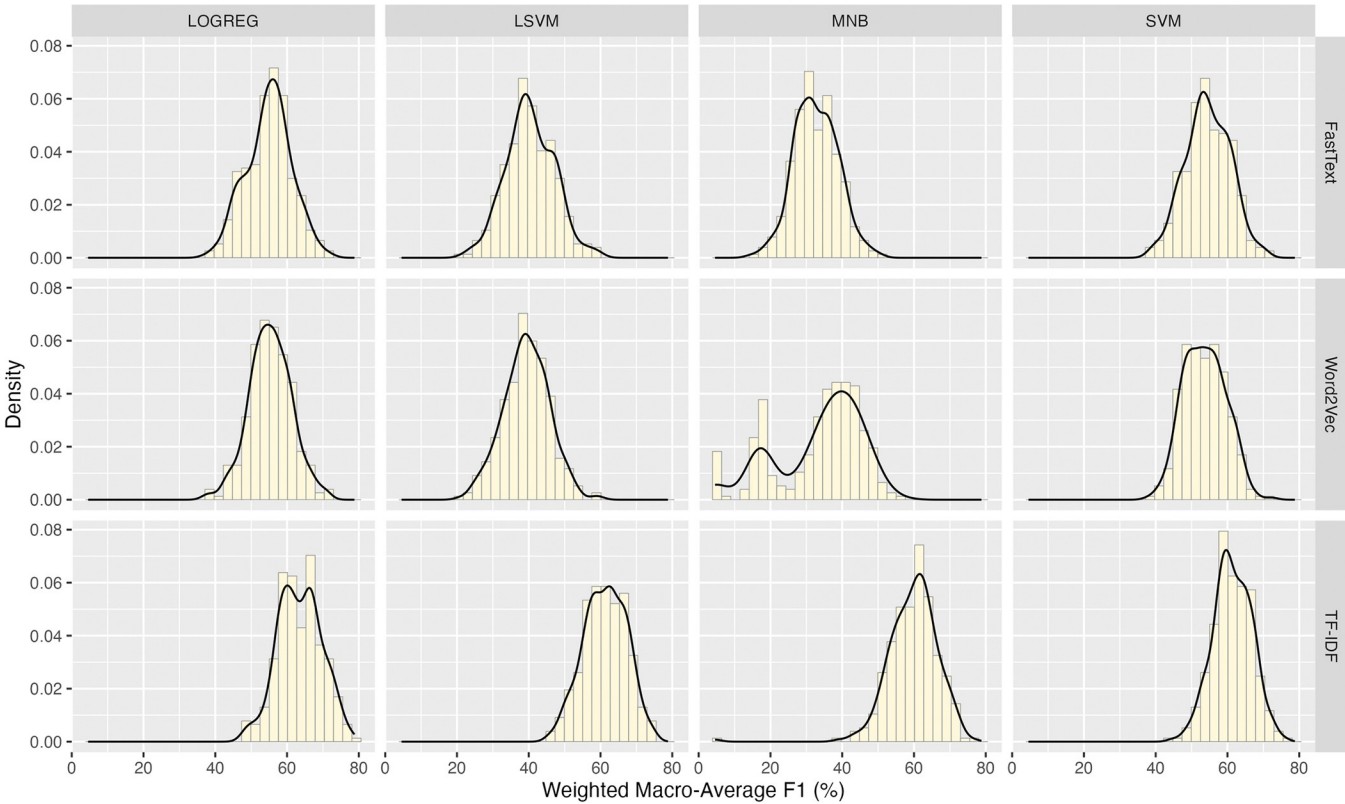

**Fig 7. The distributions of the weighted macro-average F1 values for groups of various combinations of feature representation schemes and machine learning models.**

## Discussion

The feature representation schema impacts the classifier's performance the most, followed by the machine learning model and the interaction between the two, according to the experiment results. The feature representation schema converts the document/question features into numerical representations for machine learning models. Poor feature representations reduce the classifier's performance. Many studies have focused on finding good feature representations to improve the classifier's performance, such as the feature representation for question classification [12, 13, 18] or document classification [41, 42]. In the experiments, the feature representation schema factors had the largest effect size, consistent with the literature.

A good feature representation schema improves the classifier's performance and consistency across different machine-learning models. As shown in the mean analysis of the interactions in Figs 2, 5, and 8, the TF-IDF scheme classifiers had better performance and smaller performance variances across different machine learning models than those with the Word2-Vec and FastText schemes. That implies that the quality of the feature representation schema should be prioritized when designing a question classifier.

When comparing the feature representation schemas, the experiment results indicated that the classifiers with the TF-IDF scheme outperformed those with the Word2Vec and FastText schemes in all performance metrics. Martinčić-Ipšić et al. [43] found that the TF-IDF method is not inferior to the word embedding method. Dessí et al. [44] and Khanna et al. [45] also reported that the TF-IDF method outperformed the word embedding scheme in document classification, especially for short documents [46].

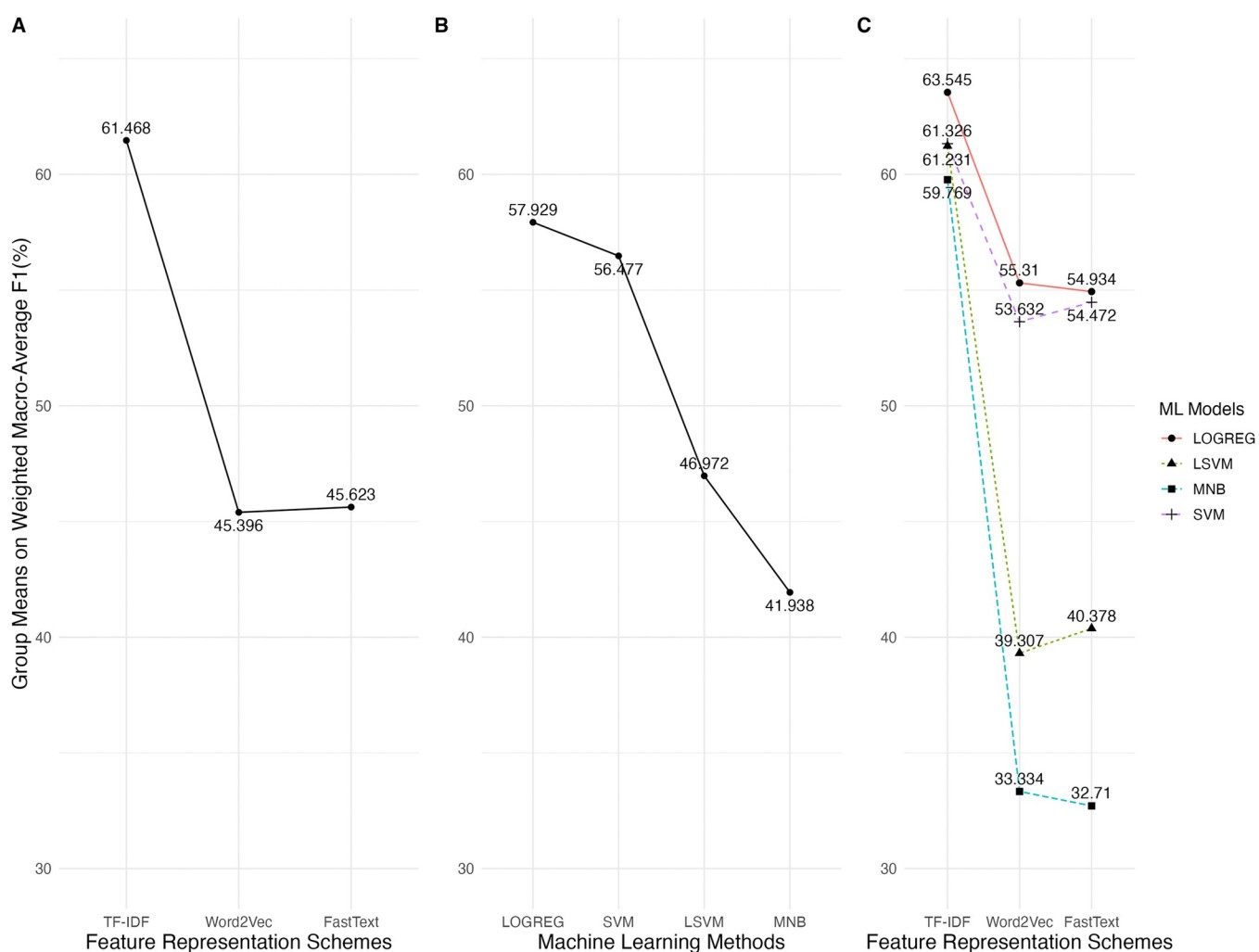

**Fig 8. The mean analysis of the weighted macro-average F1 values for FRS, MLM factors, and their interactions.**

There might be two reasons for the inferior performance of the word embedding scheme in question classification. Firstly, although the word embedding scheme captures the semantics of words, it does not consider the importance of words in the feature representation [47]. Secondly, the pre-trained word embedding schemes used in the study might need to learn the vocabulary in the SQL syntax or the table or column names. The table or column names in the SQL syntax are often abbreviated or combined with multiple words to form new names. These abbreviations and the new names might cause out-of-vocabulary problems in the pre-trained word embedding schemes. As a result, these schemes might lose some domain knowledge and harm the feature representation quality.

Nevertheless, the word embedding scheme sometimes outperforms the TF-IDF scheme. Arora et al. [48] reported that the word embedding scheme outperformed the TF-IDF scheme in classifying the instances in the TagMyNews dataset. They trained its word embeddings instead of pre-trained ones. That implies that if word embeddings can effectively represent the semantic information in the knowledge domain, word embedding would be more effective than TF-IDF.

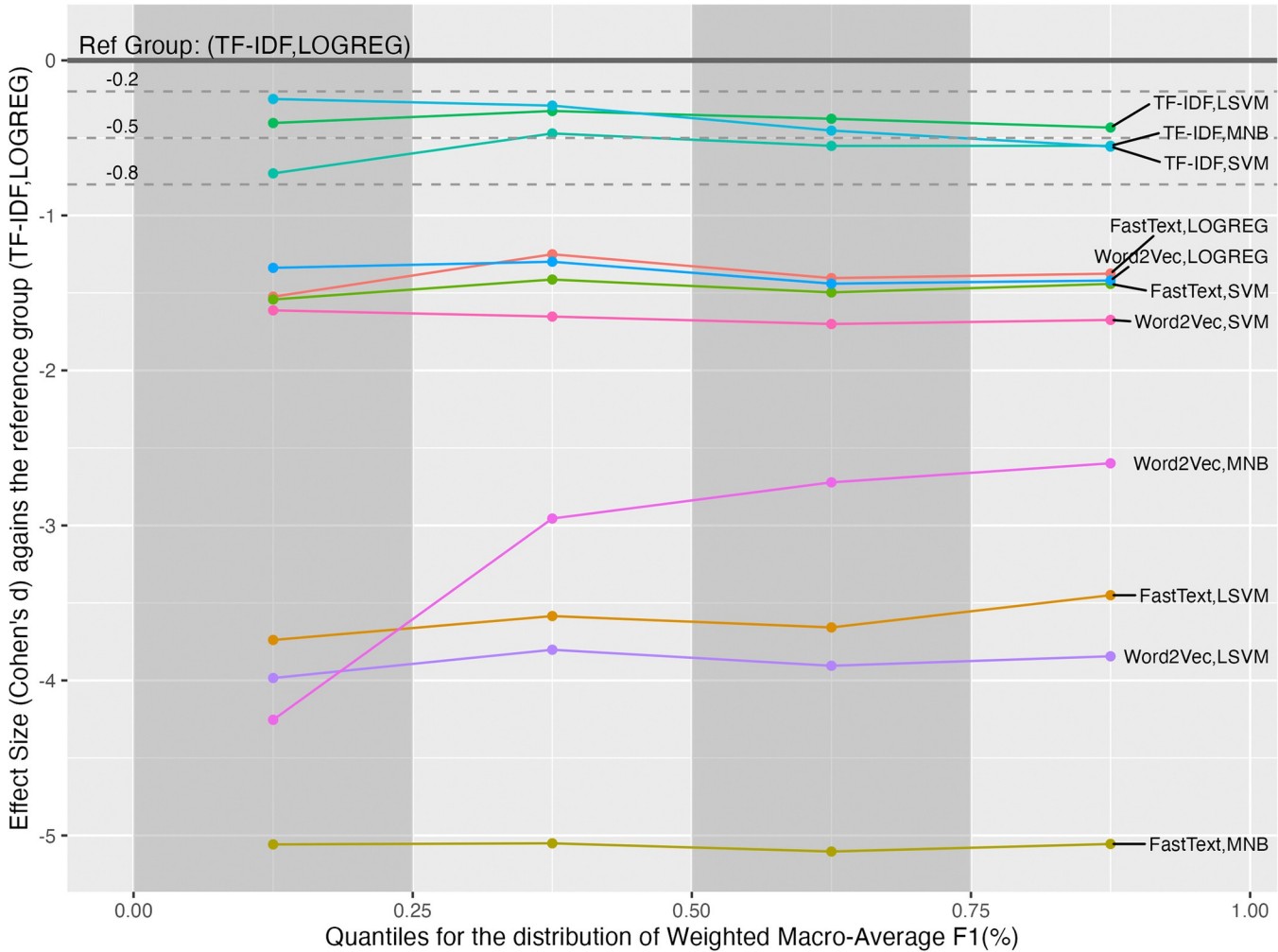

**Fig 9. The effect size analysis on the weighted macro-average F1 values in the four quantiles for the interactions between the FRS and MLM factors.**

As for the machine learning models, no consensus is reached on which model performs the best in all performance metrics in the experiment. The logistic regression model outperformed the others in the weighted AUC and weighted F1 metrics; the linear support vector machine outperformed the others in the weighted precision metric, as indicated by the experiment results.

Many studies in question classification reported that the SVM model outperformed the others [12, 13, 15, 18, 23]. These studies mainly used the TF-IDF scheme. In the study's experiment, when using the TF-IDF scheme, the logistic regression model outperformed the others in the weighted AUC and weighted F1 metrics, and so did the SVM model in the weighted precision metric. Our experiment results are partially consistent with the literature.

Does the SVM model outperform the logistic regression model in all cases? No consensus exists in the literature. Pranckevičius and Marcinkevičius [49] reported that the logistic regression model outperformed the SVM model in classifying text reviews using the TF-IDF scheme. Musa [50] reported that the logistic regression model outperformed the SVM model in predicting the relevance of heart disease.

Theoretically, the loss functions of the SVM and logistic regression models behave similarly. Hence, they should have similar performance. If the separation between class instances is

clear, the SVM model would outperform the logistic regression model. On the other hand, if the class instances overlap, the logistic regression model would outperform the SVM model [34]. Empirically, Salazar et al. [51] reported that the data distribution impacts the performance of the two models. If the data distribution is univariate, the logistic regression model outperforms the SVM model. Otherwise, the SVM model outperforms the logistic regression model. The data distributions in all groups in the experiment were univariate. Hence, it is reasonable that the logistic regression model sometimes outperformed the SVM model.

To summarize the discussion, the factors that impact the performance of classifiers include (i) the data distribution, (ii) the quality of the feature representation schema, (iii) the machine learning model, and (iv) the parameter settings. Hence, one should first consider the feature representation schema. Then, choose the appropriate machine learning model, considering the data distribution. Finally, tune the parameter settings of the machine learning model to achieve the best performance.

## Conclusion

The study explores factors that affect the design of automatic classifiers for questions containing code snippets through factorial experiments, taking Oracle SQL certification exam questions as examples. Our research results show that the TF-IDF and Logistics Regression classifier performed best in the weighted macro-average AUC and weighted macro-average F1; the classifier with TF-IDF and Support Vector Machine performed best in the weighted macro-average Precision. Besides, the experiment results indicate that the feature representation scheme produces a more significant effect size than the machine learning method on the performance of the question classifiers. A good feature representation scheme can improve the performance and consistency of different machine learning methods. In addition, logistic regression and SVM models performed better than the linear SVM and MNB models.

Based on the experiment results and literature, this study concludes that the data distribution, the quality of the feature representation scheme, the machine learning models, and the parameter settings of the models are the main factors affecting the performance of the classifiers for questions containing code snippets. The feature representation scheme should be the first factor when designing a question classifier. Upon deciding on the feature representation scheme, one then chooses the appropriate machine learning model, considering the data distribution. Finally, fine-tune the parameter settings of the machine learning model to achieve the best performance.

The contributions of this study are twofold. First, the study explores the design of automatic classifiers for questions containing code snippets. Hence, it fills the gap in the literature on question classification, which mainly focuses on questions with general text descriptions. Second, the study's results enable teachers/practitioners to build a question classification system to suggest a topic for a question. That can help them automatically classify the collected historical questions to accelerate building a test bank and reduce teachers'/practitioners' workload.

Due to the limited research resources, the study has the following limitations: (1) the number of questions in the dataset is 171. If the number of questions increases, other advanced models, such as neural networks or deep learning models, may be employed; (2) this study uses pre-trained word embedding schemes. The results might differ if self-trained word embedding schemes are used; (3) this study mainly discusses four machine learning models: logistic regression, SVM, linear SVM, and Multi-nominal Naive Bayes. Other machine learning models or ensemble learning methods may produce different results; (4) this study only focuses on SQL certification exam questions. Whether one can directly extend our research results to questions containing code snippets in other programming languages (such as Java, Python, etc.) needs further research.

In the future, one can develop ensemble learning models to improve the performance of the classifiers. Alternatively, one can establish classifiers with the incremental learning method to learn new instances online to adapt to the pattern changes when adding new questions to the test bank.

## Supporting information

**S1 Appendix. Appendix nomenclatures to express the statistical results.**
(DOCX)

## Author Contributions

**Conceptualization:** Hung-Yi Chen.

**Formal analysis:** Hung-Yi Chen, Po-Chou Shih.

**Methodology:** Hung-Yi Chen, Yunsen Wang.

**Software:** Hung-Yi Chen, Po-Chou Shih.

**Validation:** Po-Chou Shih.

**Visualization:** Hung-Yi Chen.

**Writing – original draft:** Hung-Yi Chen.

**Writing – review & editing:** Po-Chou Shih, Yunsen Wang.

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
