## [Decision Letter · Decision Letter 0]

2 May 2024

PONE-D-24-07100Exploration of Designing an Automatic Classifier for Questions Containing Code Snippets - A Case Study of Oracle SQL Certification Exam QuestionsPLOS ONE

Dear Dr. Shih,

Thank you for submitting your manuscript to PLOS ONE. After careful consideration, we feel that it has merit but does not fully meet PLOS ONE’s publication criteria as it currently stands. Therefore, we invite you to submit a revised version of the manuscript that addresses the points raised during the review process.

We look forward to receiving your revised manuscript.

Kind regards,

Vincenzo Bonnici, PhD

Academic Editor

PLOS ONE

Reviewers' comments:

Reviewer's Responses to Questions

**Comments to the Author**

1. Is the manuscript technically sound, and do the data support the conclusions?

Reviewer #1: Yes

2. Has the statistical analysis been performed appropriately and rigorously? 

Reviewer #1: Yes

3. Have the authors made all data underlying the findings in their manuscript fully available?

Reviewer #1: Yes

4. Is the manuscript presented in an intelligible fashion and written in standard English?

Reviewer #1: Yes

5. Review Comments to the Author

Reviewer #1: The authors evaluate the factors that affect the design of a classifier for questions containing SQL code snippets using the public dataset from the Oracle SQL certification exam as benchmark.

Comments:

- Line 47. “Bloom Taxonomy” A definition and/or explanation can help the reader.

- L88. The Literature Review section is very poor and poorly edited, unless considering the following sections, it seems more like a piece of the Introduction. Perhaps it would be better to integrate the following sections (L97, L113, and L131) into this one.

- TF-IDF/TFPOS-IDF and subsequent. Acronyms must be introduced before using them.

- L97. “Training Data Set” Perhaps, is "Classification Schemes" a more appropriate title?

- L99. “a data set in a domain is challenging to apply to another domain” Is this transfer learning or fine-tuning?

- Table 1. The alignment of the lines is unclear and makes reading complicated.

- L118. TFPOS-IDF needs a short explanation. The same for ETF-IDF and ETFPOS-IDF.

- L127. Is E-TFIDF a typos or a new method?

- L166. This section title does not seem necessary.

- L192. “manually labeled the questions” Is that the second column of Table 4?

- L221-223. These three lines should be better contextualised.

- L234. “We generate the question embeddings by compositing the word embeddings.” How? This part needs a better explanation.

- L235. “Let vij , i = 1 ... nj denote the” does question j have exactly j words?

- L287. “as follows: The recall” A capital letter should not be used after a colon.

- L289. “instances from TP and FN instances” Acronyms must be introduced before using them.

- L292. In equation (1), what does c represent?

- L361. “the following nomenclatures” Consider whether to move it to a separate section or to the appendix.

- Section Experiment Results and Discussion. The figures are of very low quality and not readable. In addition, tables could be added to better summarise the results.

- L536-540. “TF-IDF scheme outperformed those with the Word2Vec and FastText schemes”, “with the TF-IDF scheme outperformed those with the Word2Vec and FastText” Unhelpful repetitions. This section should be better maintained.

- From L545 to L553. Can these results also be valid with applications that contain code from other languages, such as C++ or Java?

- L556. “classifying the instances in the TagMyNews dataset.” It seems quite obvious since it is a different context. Perhaps it should be better explained.

- Check references section. For instance, who is the author-editor of the first entry?

6. PLOS authors have the option to publish the peer review history of their article (what does this mean?). If published, this will include your full peer review and any attached files.

Reviewer #1: No

---

## [Author Response · Author response to Decision Letter 0]

14 May 2024

The authors sincerely appreciate the reviewers' valuable comments and suggestions for improving the manuscript's quality. We have provided the Response-to-Reviewer letter, which responds to the reviewer's comments point-by-point. The letter is attached as a file in the re-submission process.

---

## [Decision Letter · Decision Letter 1]

6 Aug 2024

Exploration of Designing an Automatic Classifier for Questions Containing Code Snippets - A Case Study of Oracle SQL Certification Exam Questions

PONE-D-24-07100R1

Dear Dr. Shih,

We’re pleased to inform you that your manuscript has been judged scientifically suitable for publication and will be formally accepted for publication once it meets all outstanding technical requirements.

Kind regards,

Vincenzo Bonnici, PhD

Academic Editor

PLOS ONE

Additional Editor Comments (optional):

Reviewers' comments:

Reviewer's Responses to Questions

**Comments to the Author**

1. If the authors have adequately addressed your comments raised in a previous round of review and you feel that this manuscript is now acceptable for publication, you may indicate that here to bypass the “Comments to the Author” section, enter your conflict of interest statement in the “Confidential to Editor” section, and submit your "Accept" recommendation.

Reviewer #1: All comments have been addressed

Reviewer #2: All comments have been addressed

2. Is the manuscript technically sound, and do the data support the conclusions?

Reviewer #1: Yes

Reviewer #2: Yes

3. Has the statistical analysis been performed appropriately and rigorously? 

Reviewer #1: Yes

Reviewer #2: Yes

4. Have the authors made all data underlying the findings in their manuscript fully available?

Reviewer #1: Yes

Reviewer #2: Yes

5. Is the manuscript presented in an intelligible fashion and written in standard English?

Reviewer #1: Yes

Reviewer #2: Yes

6. Review Comments to the Author

Reviewer #1: (No Response)

Reviewer #2: The comments are appropriately addressed. The grammar of the writeup was improved, compared with the previous version. There are no changes.

7. PLOS authors have the option to publish the peer review history of their article (what does this mean?). If published, this will include your full peer review and any attached files.

Reviewer #1: No

Reviewer #2: **Yes: **Ramesh Kumar Ayyasamy

---

## [Editor Report · Acceptance letter]

23 Aug 2024

PONE-D-24-07100R1 

PLOS ONE

Dear Dr. Shih, 

I'm pleased to inform you that your manuscript has been deemed suitable for publication in PLOS ONE. Congratulations! Your manuscript is now being handed over to our production team.

Kind regards, 

on behalf of

Dr. Vincenzo Bonnici 

Academic Editor

PLOS ONE